# REINFORCEMENT LEARNING BASED IMAGE GENERATION VIA VISUAL CONSENSUS EVALUATION

## ABSTRACT

Image generation models are typically trained using the L2 or cross-entropy loss, and evaluated using IS or FID. The inconsistency between the training and evaluation metrics results in suboptimal model performance. To this end, we explore to address the aforementioned issue by finetuning pre-trained generative models with the reinforcement learning. Considering that current evaluation metrics can not be used as training objects since obtaining an accurate score typically demands more than ten thousand images, we introduce an innovative automated metric that captures consensus as a reward signal of the reinforcement learning for finetuning image generation models. It exhibits strong correlation with commonly used metrics such as FID, and demonstrates better robustness to the number of images than FID. Experiments indicate that when introducing varying degrees of noise to the generated images, such as ImageNet contamination or Gaussian noise, our metric quantifies the level of disruption more accurately than IS. By finetuning generative models with our proposed method, we boost the performance for image generation on multiple benchmarks like LSUN 256x256 and ImageNet 64x64.

## 1 INTRODUCTION

In recent years, image generation models have achieved remarkable progress and demonstrated their ability to produce realistic and diverse visual content. These models are typically trained using loss functions such as L2 or cross-entropy lossand evaluated using metrics like Inception Score (IS) Barratt & Sharma (2018) or Fréchet Inception Distance (FID) Heusel et al. (2017). However, considering that the metrics employed during training is inconsistent with that used for evaluation, current methods generate images with suboptimal quality Ranzato et al. (2015). It is since that traditional training losses can not properly instruct models to capture the intricacies of image quality and diversity, thus damaging model performance.

Despite that the evaluation metrics, such as IS and FID, can provide valuable insights into the quality and diversity of generated images, they require a substantial number of samples to produce reliable scores. In particular, accurately assessing image quality with IS and FID typically necessitates a large number of images, often exceeding tens of thousands. Moreover, the commonly used back-propagation training paradigm can not be applied to these evaluation metrics since they are non-differentiable. In the end, it is difficult to use these evaluation metrics for guiding the training process of image generation models.

In this paper, we propose a novel metric that is strong-correlated with previous evaluation metrics but requires much smaller number of generated images. To optimize the image generation models under the direct guidance of non-differentiable metrics, we propose a novel approach to leverage reinforcement learning (RL). By fine-tuning pre-trained generative models using our proposed metric through reinforcement learning, we bridge the gap between the training and evaluation phases, thus improving image generation performance. In particuler, to reduce the number of images required for evaluation, we introduce an innovative automated visual consensus-based Evaluation metric as a reward signal for reinforcement learning based image generation. Specifically, for a pair of reference and generated images forming an image pair, we encode the images into two token sequences using a pre-trained VQ-VAEVan Den Oord et al. (2017). Subsequently, we compute the TF-IDF vectors for each $nxn$ contiguous tokens combination, followed by utilizing cosine similarity to measure the semantic consistency between the reference and generated images. This new metric offers an alter-

native approach to quantifying image quality and diversity, which can capture nuanced aspects that are vital for effective training.

Extensive experiments demonstrate the effectiveness and efficiency of our method. Notably, when introducing perturbations such as contamination of the LSUN bedroom dataset with ImageNet images or Gaussian noise to the generated images, our metric outperforms the Inception Score in accurately quantifying the level of disruption. This emphasizes the ability of our metric to discern subtle variations in image quality. Moreover, our proposed automated metric demonstrates a robust correlation with widely-used metrics like FID while displaying enhanced stability across varying sample sizes. Furthermore, unlike IS and FID, which exhibits sensitivity to the number of images used for evaluation, our metric showcases a higher level of consistency, making it more suitable for reinforcement learning based finetuning.

Through the application of our proposed method, we observe significant performance enhancements in image generation across multiple benchmark datasets, including LSUN 256x256Yu et al. (2015), and ImageNet 64x64Deng et al. (2009). This demonstrates the potential of reinforcement learning based image generation via our CIGE metric in pushing the boundaries of image generation capabilities.

Our contributions can be summarized as follows:

- We introduce a Consensus-based Evaluation metric for Image Generation that can accurately identify noise and exhibits strong robustness with respect to the number of images.
- We introduce reinforcement learning to image generation models to directly optimize non-differentiable metric, thereby avoiding the inconsistency between the training and evaluation metrics.
- We assess the effectiveness of our methods on LSUN 256x256 and ImageNet 64x64 datasets. The experiments demonstrate the efficacy of our approach.

## 2 RELATED WORK

**Consensus-based Image Desciption Evaluation** Text generation evaluation metrics primarily include BLEU Papineni et al. (2002), ROUGE Lin (2004), METEOR Banerjee & Lavie (2005), and CIDEr Vedantam et al. (2015). CIDEr emphasizes consensus-based evaluation, treating alignment between generated descriptions and multiple reference descriptions as a critical criterion. This approach aligns more closely with human perception. Given an image and a collection of human generated reference sentences describing it, the objective of CIDEr is to measure the similarity of a candidate sentence to a set of reference sentences. The measure of consensus involves assessing how frequently the n-grams in the candidate sentence are present in the reference sentences. Conversely, n-grams absent from the reference sentences should not be present in the candidate sentence. Furthermore, n-grams that exhibit commonality across the entire dataset should be assigned diminished weight due to their presumed lower informativeness. We believe that the core concept of CIDEr can be applied to the evaluation of image generation tasks.

**Evaluation Metrics of Image Generation** In image generation tasks, the most commonly used evaluation metrics are IS and FID. The Inception Score assesses the diversity and realism of generated images by computing a model's performance. It employs a pre-trained image classifier, Inception-V3 Szegedy et al. (2016), to calculate the class probability distribution for each generated image, then measures diversity and quality using KL divergence. However, since Inception-V3 is trained on the ImageNet dataset, the Inception Score is biased towards ImageNet characteristics, considering anything dissimilar to ImageNet as less authentic. Inception Score heavily relies on the classifier and indirectly evaluates image quality, neglecting specific differences between real and generated data.

FID evaluates the similarity between the distributions of generated and real images by computing the Fréchet distance. It measures the dissimilarity between two multivariate normal distributions, where lower values are preferable. However, FID is highly sensitive to the number of images and demands a significant amount of data to be computed effectively.

**Reinforcement Learning based Generation Models** In NLP, text generation models are typically based on n-grams Kneser & Ney (1995), feed-forward neural networks Morin & Bengio (2005), re-

current neural networks Mikolov et al. (2010) or transformers Vaswani et al. (2017). They are trained to predict the next word given the preceding real words as input. During testing, the trained model is used to sequentially generate a sequence, taking the generated words as input. Errors accumulate along the way, leading to exposure bias issues. Furthermore, these models are trained with word-level losses (e.g., cross-entropy) to maximize the probability of the next word. However, they are evaluated based on different metrics such as BLEU. To address these issues, we apply reinforcement learning to text generation. In 2016, Mixed Incremental Cross-Entropy Reinforcement Ranzato et al. (2015) was introduced for sequence prediction. It employs incremental learning and combines the REINFORCE Sutton & Barto (2018) algorithm with cross-entropy loss. MIXER is a sequence-level training algorithm that adjusts the training and testing objectives, such as BLEU, rather than predicting the next word as in previous works. The actor-critic algorithm is applied to sequence prediction, aiming to further enhance MIXER. It utilizes a critic network to predict the value of tokens, which represents the expected score following the sequence prediction policy defined by the actor network, and is trained using the predicted token values Bahdanau et al. (2016).SeqGAN Yu et al. (2017) is a sequence generative adversarial network with policy gradients, integrating adversarial training from GANs Goodfellow et al. (2020). ChatGPT Ouyang et al. (2022) employs reinforcement learning fine-tuning based on human feedback to finetune pre-trained language model.

In the field of image generation, there is still limited research available. Recently, ImageReward Xu et al. (2023) attempted to enhance text-to-image models through human preference feedback. However, it only presents evaluation results using the evaluation method proposed in the paper itself and human evaluation.

**Consistency Models** Recently, diffusion model based image generation models have seen rapid development. These models transform Gaussian noise into samples from a known data distribution through iterative denoising processes Ho et al. (2020). The generated images exhibit notable diversity and realism However, methods based on diffusion model often require multiple iterations during the inference process, which affects the generation speed. Recently, consistency models Song et al. (2023) have addressed this issue. It supports fast one-step generation by design, while still permitting a few-step sampling approach to balance computational cost and sample quality. In this paper, we employ consistency models as the pre-trained framework.

## 3 METHOD

In this section, we introduce Consensus-based evaluation metric for Image Generation and the reinforcement learning based consistency models.

### 3.1 CIGE METRIC

We devise an automated metric that captures consensus. Our goal is to automatically evaluate for image $I_i$ how well a generated image $g_i$ matches the reference image $r_i$. First, we train a VQ-VAE with added perceptual loss Johnson et al. (2016), and the encoder of this VQ-VAE serves as an image tokenizer to transform an image to a sequence of discrete tokens. We consider a contiguous block of $nxn$ tokens as a single entity in spatial proximity. CIGE measures consensus by encoding how often $nxn$-tokens in the generated image are present in the reference image. While $nxn$-tokens not present in the reference image should not be in the generated image. Moreover, $nxn$-tokens that commonly occur across all images in the dataset should be given lower weight, since they are likely to be less informative. In particular, we perform a TF-IDF **?** weighting for each $nxn$-tokens. The number of times an $nxn$-tokens $\omega_k$ occurs in a reference image $r_i$ is denoted by $h_k(r_i)$ or $h_k(g_i)$ for the generated sentence $g_i$. We compute the TF-IDF weighting $g_k(s_i)$ for each $nxn$ token $\omega_k$ using:

$$g_k(s_i) = \frac{h_k(s_i)}{\sum_{\omega_l \in \Omega} h_l} \log \left( \frac{|I|}{\sum_{I_p \in I} \min(1, \sum_q h_k(s_q))} \right) \tag{1}$$

where $\Omega$ is the set of all $nxn$ tokens and I is the set of all images in the dataset. The first term measures the TF of each $nxn$ token $\omega_k$, and the second term measures the rarity of $\omega_k$ using its IDF. Intuitively, TF places higher weight on $nxn$ tokens that frequently occur in the reference image, while IDF reduces the weight of $nxn$ tokens that commonly occur across all images in the dataset. That is, the IDF provides a measure of word saliency by discounting popular tokens that are likely

to be less visually informative. The IDF is computed using the logarithm of the number of images in the dataset $|I|$ divided by the number of images for which $\omega_k$ occurs in any of its reference images.

Our $\text{CIGE}_n$ score for $n\mathrm{x}n$ tokens of length n is computed using the average cosine similarity between the generated image and the reference image, which accounts for both precision and recall:

$$\text{CIGE}_n(g_i, r_i) = \frac{1}{m}\frac{\boldsymbol{g}^n(c_i) \cdot \boldsymbol{g}^n(r_i)}{\|\boldsymbol{g}^n(c_i)\|\|\boldsymbol{g}^n(r_i)\|}) \tag{2}$$

where $\mathbf{g^n}(c_i)$ is a vector formed by $g^k(c_i)$ corresponding to all $n\mathrm{x}n$ tokens of length n and $\|g^n(c_i)\|$ is the magnitude of the vector $g^n(c_i)$. Similarly for $g^n(r_i)$. In the experiments conducted in this paper, we set $n = 2$.

## 3.2 FINETUNE CONSISTENCY MODELS WITH THE REINFORCEMENT LEARNING

The learning algorithms we describe in the following sections are agnostic to the choice of the underlying model, as long as it is parametric. In this work, we focus on diffusion models as they are a popular choice for image generation. In particular, we use consistency models.

### 3.2.1 PRETRAIN CONSISTENCY MODELS

We first train a consistency model as a pretraining step. A prominent characteristic of the consistency model is self-consistency: points along the same PF ODE trajectory map to the same initial point Song et al. (2023).

When training consistency models, we utilize numerical ODE solvers and a pre-trained diffusion model to generate pairs of adjacent points on a PF ODE trajectory. Consider discretizing the time horizon $[\epsilon, T]$ into $N - 1$ sub-intervals, with boundaries $t_1 = \epsilon < t_2 < \cdots < t_N = T$. When $N$ is sufficiently large, we can obtain an accurate estimate of $x_{t_n}$ from $x_{t_{n+1}}$ by running one discretization step of a numerical ODE solver. This estimate, which we denote as $\hat{x}_{t_n}^\phi$, is defined by

$$\hat{x}_{t_n}^\phi := x_{t_{n+1}}^\phi + (t_n - t_{n+1})\Phi(x_{t_{n+1}}^\phi, t_{n+1}; \phi), \tag{3}$$

where $\Phi(\cdots; \phi)$ represents the update function of a one-step ODE solver applied to the empirical PF ODE.

By minimizing the discrepancy between model outputs for these pairs, we can effectively transform a diffusion model into a consistency model. This enables the generation of high-quality samples with just a single network evaluation. The consistency distillation loss is defined as

$$\mathcal{L}_{pre} = \mathbb{E}[\lambda(t_n)d(f_\theta(x_{t_{n+1}}, t_{n+1}), f_{\theta^-}(\hat{x}_{t_n}^\phi, t_n)], \tag{4}$$

where $\lambda(\cdot) \in \mathbb{R}^+$ is a positive weighting function, $\hat{x}_{t_n}^\phi$ is given by Eq. (4), $\theta^-$ denotes a running average of the past values of $\theta$ during the course of optimization, and $d(\cdot, \cdot)$ is a metric function that satisfies $\forall x, y : d(x, y) \geq 0$ and $d(x, y) = 0$ if and only if $x = y$.

### 3.2.2 FINETUNE WITH REINFORCEMENT LEARNING

As described in the previous section, image generation systems are traditionally trained using the L2 or cross entropy loss. To directly optimize metrics, we can cast our generative models in the Reinforcement Learning terminology. Generative models can be viewed as an "agent" that interacts with an external "environment" (image features). The parameters of the network, $\theta$, define a policy $p_\theta$, that results in an "action" that is the prediction of the image. For each step during training, the agent observes a "reward" that is, for instance, the CIGE score of the generated image——we denote this reward by $R$. The reward is computed by CIGE metric by comparing the generated image to corresponding ground-truth image. The goal of training is to minimize the negative expected reward:

$$\mathcal{L}_\theta = -\mathbb{E}_{x_i \sim I}(R(g_i, r_i)), \tag{5}$$

where $g_i$ is the generated image at step i, and $r_i$ is the reference image. We directly fine-tune consistency models by viewing the CIGE score as the losses.

$$\mathcal{L}_{reward} = -\mathbb{E}_{x_p \sim I}(R(f_\theta(x_{pt_{n+1}}), x_p) - b_p), \tag{6}$$

$$b_p = \frac{1}{N-1} \sum_{q \neq p} R(f_\theta(x_{qt_{n+1}}), x_q), \tag{7}$$

where $\theta$ denotes the parameters of the consistency models, $f_\theta(x_{qt_{n+1}})$ denotes the generated image of consistency models with parameters $\theta$. The final loss form is:

$$\mathcal{L} = \mathcal{L}_{reward} + \mathcal{L}_{pre} \tag{8}$$

## 4 EXPERIMENTS

In this section, we first demonstrate the capability of our metric to estimate the level of noise interference, its robustness with respect to the number of images, and its correlation with FID. Next, we compare the quantitative and qualitative performance of reinforcement learning-based consistency models and previous methods on LSUN Bedroom 256x256 and ImageNet 64x64 datasets. Then, we present the results of ablation studies to demonstrate the necessity of reinforcement learning.

### 4.1 METRIC

#### 4.1.1 INCREASING DISTURBANCE

We demonstrate that CIGE can accurately assess disruption levels in generated images under varying degrees of noise. Experiments indicate that our metric consistently correlates with noise levels more accurately than the Inception Score.

Fig. 5 in appendix shows that the CIGE is consistent with increasing disturbances and human judgment on the LSUN bedroom dataset. It illustrates the evaluation of CIGE and IS across different disturbances: Gaussian noise, Gaussian blur, implanted black rectangles, swirled images, salt and pepper noise, and contamination of the LSUN Bedroom dataset with ImageNet images. Remarkably, unlike the IS, the CIGE effectively captures the degree of disruption.

As shown in Fig. 5, with increasing disturbance levels, the CIGE score decreases. A higher CIGE score indicates better image quality, similar to IS. However, as the disturbance level of Gaussian noise, Gaussian blur, and ImageNet contamination increases, the IS score keeps increasing. With the continuous enhancement of the other three types of noise, the IS score remains relatively stable or exhibits fluctuations, demonstrating that it also struggles to effectively identify the other three types of noise.

We can see that IS is incapable of discerning noise presence in images. Particularly, when some images are replaced with ones from the ImageNet dataset, IS exhibits a significant increase. This is attributed to the fact that the Inception-V3 model used by IS is trained on the ImageNet dataset, which significantly impacts the accuracy of IS.

Additionally, when parts of the image undergo a spiral transformation, resembling a swirl, the decrease in CIGE score is slower compared to the addition of other types of noise. This observation aligns with human judgment.

#### 4.1.2 CONSISTENCY WITH FID

We next show that CIGE exhibits strong correlation with commonly used metrics such as FID and CLIPScore. We trained a cogview model and utilized models with varying training steps to generate 10,000 images based on MS COCO captions. Then, we calculate CIGE, FID, and CLIPScore for the generated images respectively. It can be observed that with an increase in training steps, FID gradually decreases, while CIGE and CLIPScore exhibit an increasing trend.

Additionally, we computed the Pearson correlation coefficient Sedgwick (2012) between FID and CIGE scores, resulting in an absolute value of 0.71, indicating a strong correlation between FID and CIGE. For detailed information, please refer to the appendix.

#### 4.1.3 ROBUSTNESS TO THE NUMBER OF IMAGES

In Fig. 2, CIGE exhibits superior resilience to changes in the number of images compared to FID. We employed the cogview model to generate 10,000 images based on the MS COCO dataset. Sub-

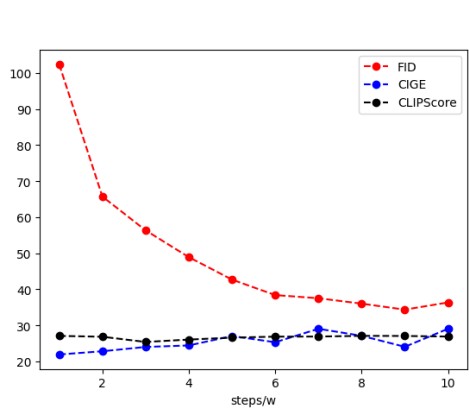

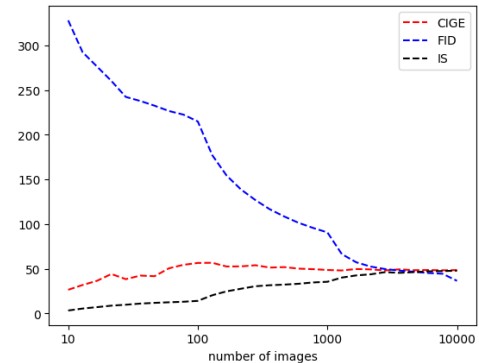

Figure 2: Robustness of metrics to the number of images. We present the curves depicting the variations of the CIGE, FID, and IS metrics as the number of images changes on MS COCO dataset.

Figure 1: CIGE exhibits strong correlation with FID .

sequently, we randomly selected N images ($10 \leq N \leq 10,000$) and computed CIGE scores, FID values, and IS scores. To better illustrate the robustness of our metrics with respect to the number of images, we applied different scaling factors to these scores to bring the CIGE score, FID, and IS values for 10,000 images closer together.

From the graph, it can be observed that as the number of images decreases, FID rapidly increases. The rate of change in IS is slower compared to the rate of change in FID, and our metric shows the least pronounced variation. When the number of images is less than 70, the change in CIGE increases in comparison to itself. This demonstrates that FID is the most sensitive to the number of images, followed by IS, whereas our metric displays the highest level of robustness.

Furthermore, even though there is some variability in the CIGE scores when calculated for individual images generated by the same model, instances of extreme values are relatively infrequent. This suggests that as the number of images reaches several tens, the average scores tend to deviate only slightly from the average scores obtained for 10,000 images. For detailed information, please refer to the appendix.

## 4.2 REINFORCEMENT LEARNING BASED CONSISTENCY MODELS

We employ consistency distillation to learn consistency models and finetuned with reinforcement learning on real image datasets. Results are compared according to Fréchet Inception Distance (lower is better), Inception Score (higher is better), Precision (higher is better), and Recall(higher is better).

### 4.2.1 BASELINES

We mainly compare our methods with diffuison models including DDPM Ho et al. (2020), EDM Karras et al. (2022), PD Salimans & Ho (2022), SS-GAN Chen et al. (2019), ADM Dhariwal & Nichol (2021).

### 4.2.2 QUANTITATIVE COMPARISON

Table 4 and Table 5 presents the results of our model trained on the LSUN Bedroom 256×256 dataset and ImageNet 64×64 dataset for image generation tasks. As shown in Table 4 and Table 5 , our approach is mainly compared with other single-step generative models.

Table 4 indicates that our method outperforms both the PD and SS-GAN approaches in terms of FID, precision, and recall when using one-step generation on the LSUN Bedroom 256×256 dataset.

| METHOD | NFE | FID | Prec. | Rec. |
|---|---|---|---|---|
| **LSUN Bedroom 256× 256** | | | | |
| DDPM | 1000 | 4.89 | 0.60 | 0.45 |
| ADM | 1000 | 1.90 | 0.66 | 0.51 |
| EDM | 79 | 3.57 | 0.66 | 0.45 |
| PD | 1 | 16.92 | 0.47 | 0.27 |
| SS-GAN | 1 | 13.3 | - | - |
| OURS | 1 | **10.74** | **0.65** | **0.30** |

Table 1: Sample quality on LSUN Bedroom 256 × 256.

| METHOD | NFE | FID | Prec. | Rec. |
|---|---|---|---|---|
| **ImageNet 64 × 64** | | | | |
| ADM | 250 | 2.07 | 0.74 | 0.63 |
| EDM | 79 | 2.44 | 0.71 | 0.67 |
| PD | 1 | 15.39 | 0.59 | 0.62 |
| OURS | 1 | **13.75** | 0.6297 | 0.6593 |

Table 2: Sample quality on ImageNet 64 × 64.

Furthermore, while the FID of our method may not be as competitive as those of DDPM, ADM and EDM, our precision is superior to that of DDPM, and it is comparable to EDM and ADM.

Table 5 indicates that our method outperforms the PD approaches in terms of FID, precision, and recall when using one-step generation on the ImageNet 64×64 dataset. Furthermore, while the FID of our method may not be as competitive as ADM and EDM, our recall is superior to that of ADM.

### 4.3 ABLATION STUDY ON THE REINFORCEMENT LEARNING

To investigate the effectiveness of our proposed finetuning with reinforcement learning, we conducted an ablation study by training consistency models without employing reinforcement learning for fine-tuning during the training process.

The quantitative results are presented in Table3, providing clear evidence of the effectiveness of our reinforcement learning approach across both datasets: LSUN Bedroom and ImageNet. Furthermore, we qualitatively compare the images generated by models using reinforcement learning for finetune and those without. We found that finetune models using reinforcement learning can better generate image detail and are less prone to distortion.

From the first three columns of images in Fig. 3, it can be observed that the shape of the bed generated by the model without fine-tuning is not close to a rectangle, and the edges of the bed are more prone to bending. The images in the fourth column indicate that our model is able to clearly generate curtains, whereas the same area in images generated by the model without fine-tuning is blurry. In the last column of images, the model without fine-tuning was not able to successfully generate the bed on the right side.

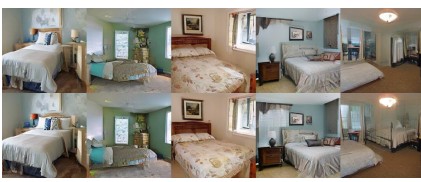

Figure 3: Images generated with(bottom) or without(top) the reinforcement learning on LSUN Bedroom 256×256. All corresponding images are generated from the same initial noise.

| Datasets | RL | FID | IS |
|---|---|---|---|
| LSUN Bedroom 256×256 | w/o | **10.70** | 2.09 |
| | w/ | 10.74 | **2.14** |
| ImageNet 64 × 64 | w/o | 13.48 | 33.44 |
| | w/ | **13.41** | **33.51** |

Table 3: Sample quality on ImageNet 64 × 64

## 5 CONCLUSION AND LIMITATIONS

This paper introduces CIGE, a visual evaluation metric that captures consensus and utilizes it as a reward signal for reinforcement learning to fine-tune consistency models.We also e introduce reinforcement learning to image generation models to directly optimize non-differentiable metric, thereby avoiding the inconsistency between the training and evaluation metrics.

Our approach offers several advantages: Firstly, when introducing varying levels of noise to the generated images, such as ImageNet contamination or Gaussian noise, CIGE provides a more accurate quantification of disruption compared to IS. Additionally, our metric exhibits a strong correlation with commonly used metrics like FID and demonstrates superior robustness to changes in the number of images compared to FID. These advantages allow us to address the disparity between training and evaluation metrics in image generation models by directly optimizing CIGE. Lastly, through reinforcement learning-based finetuning of consistency models, we enhance the model's capability to generate images.

However, our research has certain limitations. Due to constraints in computational resources, we have yet to explore the role of reinforcement learning in fine-tuning other types of generative models. Future works are encouraged to solve the above issues.

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

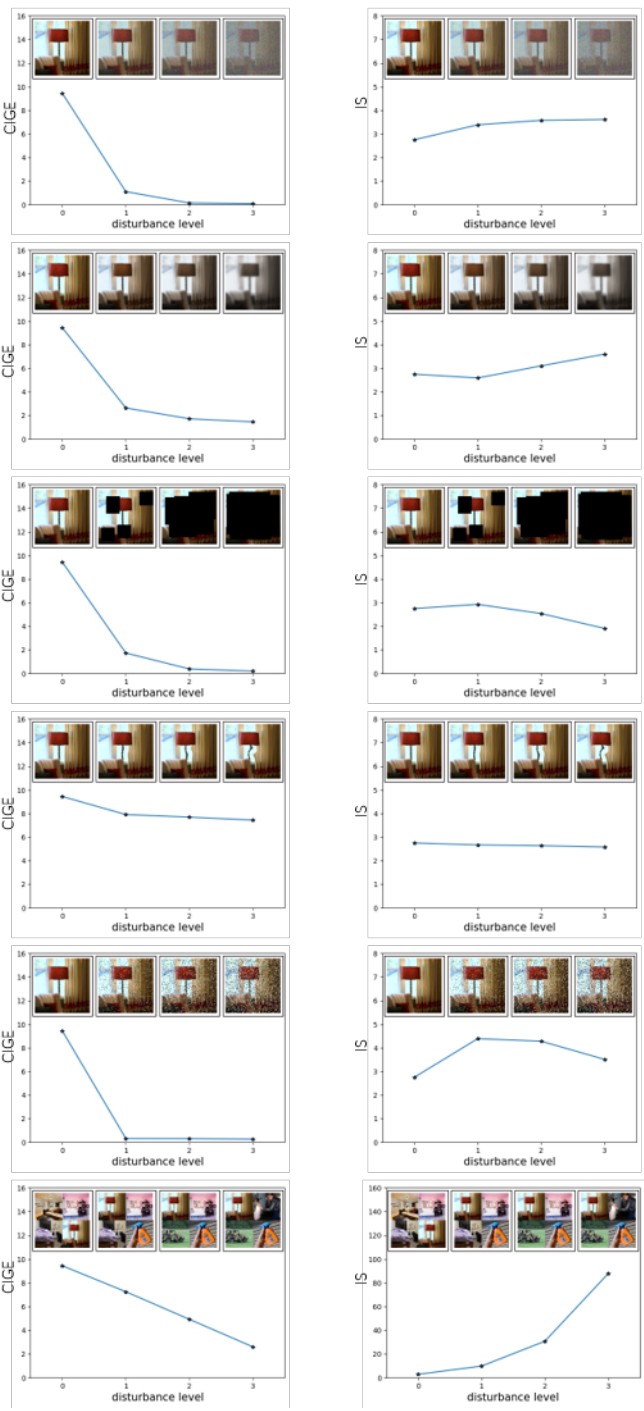

Figure 4: **Left:** CIGE and **right:** Inception Score are evaluated for **first row:** Gaussian noise, **second row:**Gaussian blur,**third row:**implanted black rectangles,**fourth row:**swirled images,**fifth row:**salt and pepper noise,**sixth row:**LSUN bedroom dataset contaminated by ImageNet images. Left is the smallest disturbance level of zero, which increases to the highest level at right. The CIGE captures the disturbance level very well by monotonically increasing whereas the Inception Score fluctuates, stay flat or even, in the worst case, decreases.

## A    APPENDIX

You may include other additional sections here.

This document provides comprehensive descriptions and results of our method that could not be accommodated in the main paper due to space restriction.

## B    ADDITIONAL EXPERIMENTS AND EXPERIMENTAL DETAILS

In this section, we present additional experiments and experimental details.

### B.1    INCREASING DISTURBANCE

We considered following disturbances of the image $X$ in section of Increasing Disturbance.

**Gaussian noise:** The noisy image is computed as $(1 - \alpha)\mathbf{X} + \alpha\mathbf{N}$ for $\alpha \in \{0, 0.25, 0.5, 0.75\}$.

**Gaussian blur:** Convolution is performed on the image using a Gaussian kernel characterized by a specific standard deviation $\alpha \in \{0, 1, 2, 4\}$.

**Black rectangles:** Add five randomly positioned black blocks to each image to obscure its original content. The size of the black blocks is determined by multiplying $\alpha$ with the image size.

**Swirl:** For a pixel with coordinates $(x, y)$ in the image, its polar coordinates relative to the center position $(x_0, y_0)$ are $(r, \theta)$, where $r = \sqrt{(x - x_0)^2 + (y - y_0)^2}$ is the radius and $\theta = \arctan((y - y_0/(x - x_0))$ is the angle. The polar coordinates after adding noise are $(r, \theta^{'})$, where $\theta^{'} = \theta - \alpha e^{-5r/(\ln 2\rho)}$. $\alpha \in \{0, 1, 2, 4\}$ is a parameter for the amount of swirl and $\rho = 25$ indicates the swirl extent in pixels. Therefore, the pixel at the original coordinates $(x, y)$ in the image corresponds to the coordinates $(x^{'}, y^{'})$ in the image after adding noise, where $x^{'} = x_0 + r\cos(\theta^{'})$ and $y^{'} = y_0 + r\sin(\theta^{'})$.

**Salt and pepper noise:** We randomly set some pixels in the image to black or white, with a probability of $50\%$ for each. The proportion of pixels that undergo changes in the image is $\alpha \in \{0, 0.1, 0.2, 0.3\}$.

**ImageNet contamination:** A percentage of $\alpha \in \{0, 0.25, 0.5, 0.75\}$ of the MSCOCO images has been replaced by ImageNet images. $\alpha = 0$ means all images are from MSCOCO, $\alpha = 0.25$ means that $75\%$ of the images are from MSCOCO and $25\%$ from ImageNet etc.

In Fig.5, we also validated the superior noise-discrimination ability of CIGE over IS and CLIPScore on the MSCOCO dataset. The CIGE captures the disturbance level very well by monotonically increasing. While the Inception Score increases when replacing some of the images with pictures from the ImageNet dataset. In addition the CLIPScore is basically consistent with increasing disturbances on the MSCOCO dataset. However, when the level of disturbance added to the images is 0 and 1, the clipscore is 29.97 and 30.03, respectively. At this point, the clipscore cannot accurately represent the quality of image generation. CLIPScore is also not sensitive to swirled distortions in images.

### B.2    PEARSON CORRELATION COEFFICIENT

The Pearson correlation coefficient, also known as Pearson's correlation, is a statistical measure used to assess the strength of a linear relationship between two variables. It quantifies the degree of linear association between two variables, with values ranging between -1 and 1, indicating the strength and direction of their correlation.

Specifically, the Pearson correlation coefficient is calculated using the following formulaSedgwick (2012):

$$r = \frac{\sum (x_i - \bar{x})(y_i - \bar{y})}{\sqrt{\sum (x_i - \bar{x})^2 \sum (y_i - \bar{y})^2}} \tag{9}$$

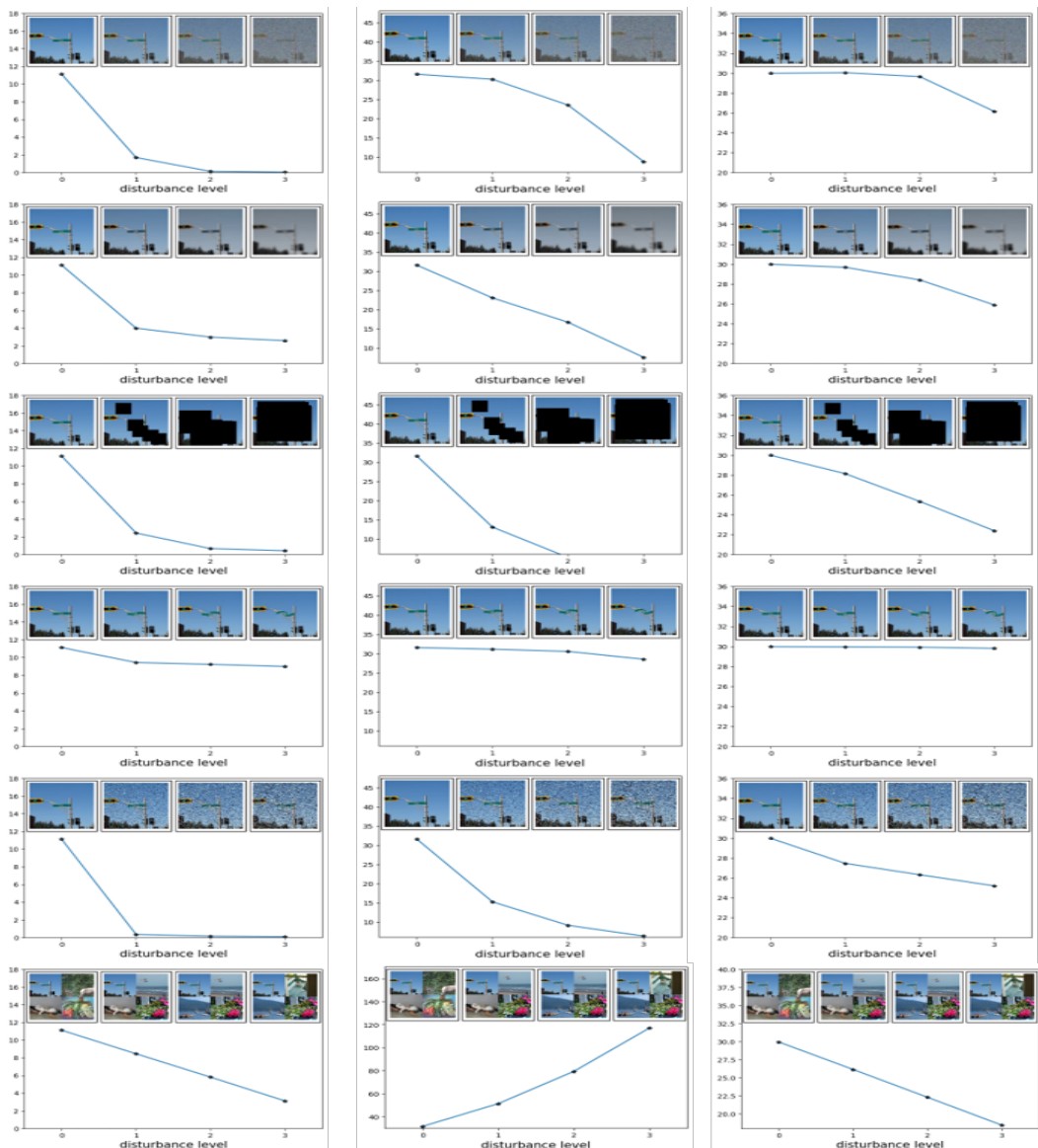

Figure 5: **Left:** CIGE, **middle:**Inception Score and **right:** CLIP Score are evaluated on MSCOCO datasets for **first row:** Gaussian noise, **second row:**Gaussian blur,**third row:**implanted black rectangles,**fourth row:**swirled images,**fifth row:**salt and pepper noise,**sixth row:**LSUN bedroom dataset contaminated by ImageNet images. Left is the smallest disturbance level of zero, which increases to the highest level at right.

1.0X¡ — X¡   ImageNet $256 \times 256$heightInput Size  $256 \times 256$Latent Layers$32 \times 32$Hidden Units128$\beta$(Commitment Loss Coefficient) 0.25Codebook Size  8192Codebook Dimension  64height

Table 4: Hyper-parameters of the VQ-VAE, the encoder of which serves as an image tokenizer to transform an image to a sequence of discrete tokens when computing CIGE.

Table 5: Hyper-parameters of the Consistency Models and reinforcement learning on LSUN Bedroom $256 \times 256$ and Imagenet $64 \times 64$.

1 p0.33 — X¡ X¡ X¡  height LSUN Bedroom $256 \times 256$  ImageNet $64 \times 64$heightBatch Size  128  32Learning Rate  1e-5  8e-

Here, $(x_i)$ and $(y_i)$ are the individual observations of the two variables, and $(\bar{x})$and $(\bar{y})$ are their respective means. This formula measures the linear relationship between the variables: when r is positive, it indicates a positive correlation, meaning that the values of both variables tend to increase or decrease together; when r is negative, it indicates a negative correlation, implying that one variable increases while the other decreases. Values of r close to 1 or -1 signify a strong linear relationship, while values near 0 suggest a weak or no linear relationship between the variables.

We trained a Cogview model and utilized models with varying training steps to generate 10,000 images based on MS COCO captions. Next, we calculate the CIGE, FID, and CLIPScore for the images generated by each of the 13 models separately. The range of variation for FID is between 31.36 and 102.36. We computed the Pearson correlation coefficient between FID and CIGE scores of these images, resulting in an absolute value of 0.71, indicating a strong correlation between FID and CIGE. Additionally, the Pearson correlation coefficient between CLIPScore and FID is -0.18.

## C    MORE QUALITATIVE RESULTS

We present more qualitative result on LSUN Bedroom $256{\times}256$ in Fig. 6.

## D    SETTINGS OF HYPER PARAMETERS

The detailed settings of model hyper parameters are presented in Table 4 and Table 5. The threshold c in Table 5 represents that when there are more than $\beta \times n \times n$ tokens in common between two $n{\times}n$-sized tokens while calculating CIGE, those two token combinations are considered as identical.

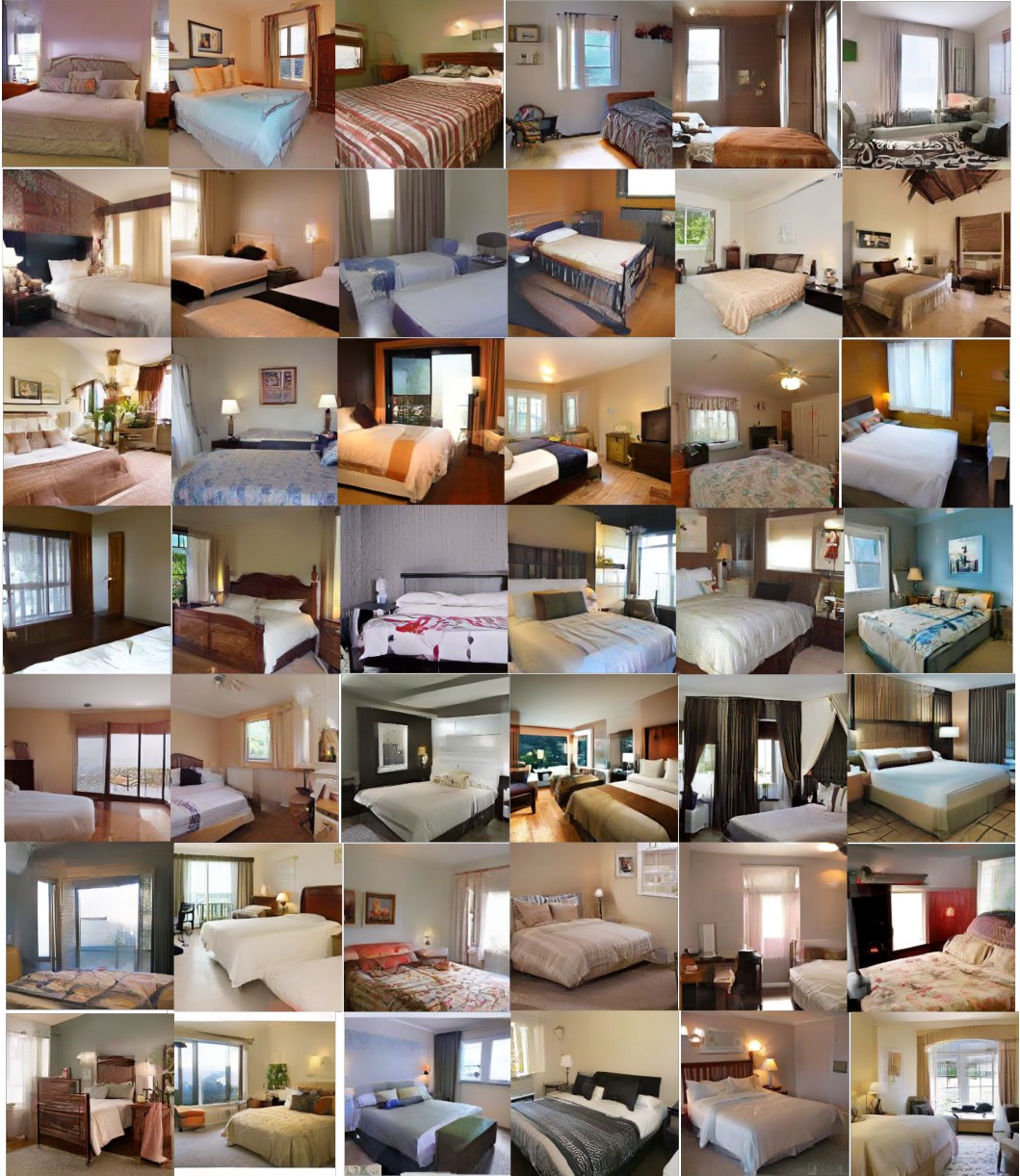

Figure 6: Qualitative result on LSUN Bedroom 256×256.

