# OpenReview forum: "Reinforcement Learning based Image Generation via Visual Consensus Evaluation"
_ICLR.cc/2024/Conference — Submitted to ICLR 2024_

### Official Review · Reviewer_Ywyw · 2023-10-30

**Soundness:** 2 fair
**Presentation:** 1 poor
**Contribution:** 2 fair
**Rating:** 1
**Confidence:** 4

**Summary:**

The paper addresses the issue of inconsistency between training and evaluation metrics in image generation models. It proposes a method that fine-tunes pre-trained generative models using reinforcement learning, with a novel automated metric that measures consensus as a reward signal. This metric correlates strongly with commonly used metrics like FID and demonstrates greater robustness to variations in the number of images. Experiments show that by applying this method, the performance of image generation models improves across various benchmarks.

**Strengths:**

The paper tries to proposes a metric that can match the human consensus of model quality with fewer number of sample unlike FID. However, the contribution is hindered by poor presentation quality.

**Weaknesses:**

1. The presentation and writing is pretty bad. For example,

- Space is not correctly used
- Some terms like abbreviation CIGE and TF-IDF come without explanation first. Better to give a discussion about TF-IDF and how it is related to your approach.
- Symbols are messy.

2. Experiments are not convincing. For example, in Table 3, how does the number support your approach?

3. Many relevant works that tried to address the issue of FID have been proposed, and they should be included in the discussion also.

Thus, the readability is pretty bad. I would suggest the authors rewriting the paper and submitting to another venue.

**Questions:**

See above.

---

### Official Review · Reviewer_6u5q · 2023-10-31

**Soundness:** 2 fair
**Presentation:** 2 fair
**Contribution:** 2 fair
**Rating:** 3
**Confidence:** 3

**Summary:**

The paper proposes a novel perceptual score following the paradigm of CIDEr for text. Thereby, a generated and a reference image are embedded using a VQ-VAE image tokenizer and the embedding sequences are treated similar as language tokens in CIDEr: generated and reference n-grams are counted and compared. In order to use the score as a loss in training generative models, the authors rely on reinforcement learning. The paper studies how the proposed score behaves under synthetic perturbations including additive noise and image distortions. The score is further applied to improve the quality of diffusion/consistency models on LSUN (256px) and ImageNet (64px).

**Strengths:**

To my knowledge the idea of using a concept similar to CIDEr to create a visual quality score is novel, and I find it pretty inspiring. Further, there seems no metric/score available with similar characteristics to FID, but which can be used as a loss (except maybe a GAN discriminator). In that sense the paper aims to tackle a highly relevant research problem.

**Weaknesses:**

Unfortunately, the paper makes a rather unpolished impression (see minor issues below), but also has some larger issues.

- Most importantly, the metric relies on a reference image for every generated image. This makes it unsuitable for unconditional generation, so it is unclear how the metric is applied to the LSUN bedroom data set (please clarify). For class-conditional ImagNet, one could use the images per class, but I’m not sure if the metric could handle the necessary diversity. Either way, details are missing on how the score is computed for the different data sets.
- According to Section 3.1 each image is mapped to 2x2 tokens. This sounds like a very low number, and the VQ-VAE would need to be extremely compressive to map a 256px image. I doubt it could meaningfully capture a lot of visual detail necessary for a good assessment of visual quality.
- According to Figure 1, the proposed “CIGE exhibits a strong correlation with FID”. This is clearly not true if the legend of that figure is correct. Further, I would expect a good visual score to capture the visual difference between FID of 100 and 30, which is quite substantial for a human.
- It seems the proposed method is not effective, despite claimed in the text: In Table 3 the FID and Inception score are almost identical with and without the proposed method/RL.


Minor issues:
- There are quite a few typos: For the first page: “training objects” -> “training objectives”, “lossand” -> “loss and”, “particuler” -> “particular”
- Notation collision for symbols in Section 3.1: g used for generated image, quantity in Eq. (1); n used for sequence length and as a superscript/exponent.
- Inception score is repeatedly computed on LSUN Bedrooms, even though it is not well suited for data other than imagenet.
- Discussion and references on Precision/Recall (Sajjadi et al. 2018) is missing from related work.

**Questions:**

Please see weaknesses for points that could be addressed in the rebuttal.

---

### Official Review · Reviewer_4fJs · 2023-11-01

**Soundness:** 2 fair
**Presentation:** 1 poor
**Contribution:** 1 poor
**Rating:** 3
**Confidence:** 4

**Summary:**

This paper present a new metric for image generative models to further improve the trained model to generate images with better quality.

**Strengths:**

1. The paper design a new training metric that is closer to the evaluation metric and then finetune the models with RL.

**Weaknesses:**

1. The paper is poor-written, hard to follow, and there are many formatting issues
 - like table 4 and table 5 referenced in sec. 4.2.2.
 - Or what is CIGE? It does not sound like Consensus-based evaluation metric for Image Generation, this should be CEIG?
 - Some reference are missed in sec. 3.1.
 - in sec. 4.1.1, it directly refers to fig. 5 in the appendix, if this figure is important, it should not be in appendix.
 - Table 1, what is NFE? for FID, lower is better, why the OURS one is bold? and which model it fineuted?
 - In sec 4.2, it mentioned IS is one of metrics but there is no IS in Table 1 and 2.
- many typos, e.g. section 5, "We also e introduce", what is the e for?

**Questions:**

1. If the proposed CIGE follows the behavior of CLIPScore, why not just use CLIPSore?
2. Is there any figure related to the description in sec. 4.1.2? Figure 1?
3. What is cogview model in 4.1.2? what is the reference?
4. For Table 3, what is number of the proposed metrics? And the difference between w/ and w/o are very small, why is that?

---

### Official Review · Reviewer_9MFC · 2023-11-01

**Soundness:** 2 fair
**Presentation:** 1 poor
**Contribution:** 2 fair
**Rating:** 3
**Confidence:** 4

**Summary:**

This paper has two contributions. 1) It introduces CIGE as a new metric, which can be used both on generative model training and evaluation. 2) It proposes a new RL-based loss term to train the consistency model. Experimental results on LSUM bedroom and Imagenet datasets show that the proposed method has better performance than the compared baselines.

**Strengths:**

- [method] CIGE shows a high correlation with the FID while being robust to fewer images, which is very interesting.
- [experiment] Experimental results on two datasets show that the proposed method has better performance than the compared baselines.

**Weaknesses:**

- [writing] This paper seems to be a rushed work and contains a lot of flaws, making it not ready for publication. Here are a few flaws I discovered: TF-IDF misses the reference, Table 3 has a wrong caption, FID in Table 1/2 is inconsistent with Table 3, Fig. 4 is placed after the reference, Fig. 4/5 are blurry, and Table 4/5 are not correctly rendered.
- [experiment] Although I like the idea of this work, especially using CIGE as a new metric on par with FID, Table 3 shows that using RL only has a marginal improvement compared with the baseline model without RL. In the LSUN bedroom dataset, using RL even makes the result worse. The above results cannot support the main claim and the effectiveness of the proposed method.

**Questions:**

No

---

### Meta-Review · Area_Chair_pQP8 · 2023-12-05

**Metareview:**

A. Paper introduces a new training metric for image generation evaluation that is better correlated with FiD
B. High correlation with FID is a nice property to have in a training metric.
C. Reviewers agree paper is poorly written/hard to follow with experiments that are not very convincing. For this reason I recommend rejection.

**Justification For Why Not Higher Score:**

poorly written + weak results

**Justification For Why Not Lower Score:**

N/A

---

### Decision · Program_Chairs · 2024-01-16

Reject